# Chemical Analysis of Various Tea Samples Concerning Volatile Compounds, Fatty Acids, Minerals and Assessment of Their Thermal Behavior

**DOI:** 10.3390/foods12163063

**Published:** 2023-08-15

**Authors:** Thomas Dippong, Oana Cadar, Melinda Haydee Kovacs, Monica Dan, Lacrimioara Senila

**Affiliations:** 1Department of Chemistry and Biology, Technical University of Cluj-Napoca, 76 Victoriei Street, 430122 Baia Mare, Romania; dippong.thomas@yahoo.ro; 2INCDO-INOE 2000, Research Institute for Analytical Instrumentation, 67 Donath Street, 400293 Cluj-Napoca, Romania; oana.cadar@icia.ro (O.C.); melinda.kovacs@icia.ro (M.H.K.); 3National Institute for Research and Development of Isotopic and Molecular Technologies, 67-103 Donath Street, 400293 Cluj-Napoca, Romania; monica.dan@itim-cj.ro

**Keywords:** tea, thermal behavior, cellulose, lignin, volatile organic compounds, fatty acids, minerals

## Abstract

Tea is the most consumed drink worldwide due to its pleasant taste and various beneficial effects on human health. This paper assesses the physicochemical analysis of different varieties of tea (leaves, flowers, and instant) after prior drying and fine grinding. The thermal decomposition behavior of the tea components shows that the tea has three stages of decomposition, depending on temperature. The first stage was attributed to the volatilization of water, while the second stage involved the degradation of volatiles, polyphenols, and fatty acids. The degradation of cellulose, hemicellulose, and lignin content occurs at the highest temperature of 400 °C in the third stage. A total of 66 volatile compounds, divided into eight classes, were identified in the tea samples. The volatile compounds were classified into nine odor classes: floral, fruity, green, sweet, chemical, woody, citrus, roasted, and alcohol. In all flower and leaf tea samples, monounsaturated (MUFAs), polyunsaturated (PUFAs), and saturated fatty acids (SFAs) were identified. A high content of omega-6 was quantified in acacia, Saint John’s Wort, rose, and yarrow, while omega-3 was found in mint, Saint John’s Wort, green, blueberry, and lavender samples. The flower and leaf tea samples studied could be a good dietary source of polyphenolic compounds, essential elements. In instant tea samples, a low quantity of polyphenols and major elements were identified. The physicochemical analysis demonstrated that both flower and leaf teas have high-quality properties when compared to instant tea.

## 1. Introduction

Tea is the world’s most commonly consumed non-alcoholic beverage next to water because of its flavor and pleasant taste and perceived health benefits such as antioxidant, antimicrobial, immunostimulatory, and antimutagenic potential, as well as its capacity to reduce cardiovascular diseases and cholesterol levels [1,2]. Tea is globally grown in as many as 58 countries on all five continents. The total land area employed for tea is estimated to be 4.12 million ha, with a production of 5.36 million tons [3]. Tea can be divided into six types or five categories, namely non-fermented tea (green), lightly fermented tea (white), semi-fermented tea (oolong), fully fermented tea (black), and post-fermented tea (yellow and dark varieties) [4]. Green, oolong, and black tea are the most widely used tea products, having different fermentation degrees and sensory profiles, with green varieties being non-fermented and black ones being fermented [5]. The tea industry is a labor-intensive sector that employs more than 13 million people worldwide [6]. The vast consumption of tea results in more than 90% of the tea left after the beverage industry [7,8]. The tea plant cultivar, terroir (climate, geography, and soil), and growing conditions are critical in establishing each tea’s aroma profile and uniqueness. Though it is possible to produce different types of tea using the same leaves, it is challenging to find a tea cultivar appropriate to obtain all six tea types of aromas and authentic profiles [6,8].

Nowadays, the instant tea market has netted a significant amount of the world market due to its ease of use and convenience in the food industry [8,9]. The production of instant tea includes oil removal, water extraction, leaf filtration, concentration, and drying. Recent studies reported that sweet instant tea has a higher content of polyphenols, flavonoids, antioxidant capacity, phlorizin, and trilobatin than initial sweet tea. As a result, instant sweet tea can be an excellent dietary source of antioxidants for consumers [9]. Extensive differences in the metabolic profiles of early and late spring tea leaves like flavonoids, flavonols, and amino acids are acknowledged, resulting in huge discrepancies in the taste and suitability of tea at different stages [10,11]. One study analyzing green teas harvested in early, middle, and late spring at low altitudes highlighted that the concentration of amino acids decreased in a gradual manner, while the concentration of carbohydrates, flavonols, and their glycosides significantly increased following late spring [10,11]. Hence, researching the accumulation of compounds in tea samples under different harvesting seasons is of great benefit for the in-depth usage of tea resources [10,11]. Tea quality is strongly correlated with catechins, caffeine, volatile compounds, and amino acids in harvestable tea shoots [12].

The elderflower flowers of *Sambucus nigra* from the *Caprofoliaceae* family are a good source of volatile oils, anthocyanins, vitamins, and phenolic compounds such as flavonoids, tannins, and coumarins. Elderflowers have a long-standing tradition in herbal medicine, displaying diuretic, laxative, antirheumatic, antiviral, and anti-inflammatory actions, leading to a strengthening of the body’s resistance and vitality [13]. Linden tea (*Tilia cordata*) is part of the *Tilinaceae* family and a highly popular herbal plant due to its beneficial properties for the central nervous system. The linden extract possesses numerous health benefits, such as being sedative, anti-inflammatory, alleviating high blood pressure, reducing anxiety, soothing digestion, and helping respiratory tract infections and insomnia [14]. *Matricaria chamomilla* is one of the most popular single-ingredient herbal teas due to its numerous health benefits, some of which include treating infections, respiratory, neuropsychiatric, gastrointestinal, and liver disorders. It is also largely used as an antispasmodic, sedative, antiemetic, and antiseptic. The main constituents of the flowers are divided into two classes: hydrophobic (terpenoids and azulenes) and hydrophilic compounds (polyphenols) [15,16]. St. John’s wort (*Hypericaceae* family) tea is the most commonly used for the treatment of water retention, insomnia, anxiety, depression, and gastritis. Its phenolic compounds include phenol carboxylic and cinnamic acids as well as flavonoids (i.e., flavones, flavanols, flavanones, flavans, catechins, bioflavonoids, and anthocyanidins). In addition to these, hypericin, pseudohypericin, prenylated phloroglucinols, and their derivatives have also been identified [17]. *Mentha piperita* and Levandula officinalis are two popular plants whose oils and teas are widely employed due to their antibacterial, antioxidant, and antidiabetic activities. Additionally, many studies support the use of Lavandula tea as a sedative, anxiolytic, and mood modulator, and Mentha tea in treating irritation and inflammation [13,18].

The volatile compounds (alcohols, aldehydes, aromatic hydrocarbons, esters, ketones, and terpenes) are principally formed during tea processing [19]. Moreover, as the main aroma substances and functional components, phenolic compounds, purine alkaloids, and free amino acids are key indicators for identifying and distinguishing the category and origin of teas [4]. Volatile compounds are easily liberated or degraded during tea processing, which can significantly affect the sensory characteristics of tea samples and the quality of the final tea product [2]. In this regard, the specific processing strongly impacts the aroma characteristics of tea, with the floral and fruity odors being mainly produced by withering or shaking and the nitrogen-containing volatiles by the combination of drying [2]. Notably, the prolonged withering for the quality formation of white tea significantly impacts the presence of volatile and non-volatile compounds, e.g., phenolic components and free amino acids, through hydrolyzation, condensation, and polymerization catalyzed by endogenous enzymes [20]. Therefore, identifying the key odorants will help understand the tea aroma chemistry and sensory markers and provide a valuable scientific basis for the different tea classifications based on the aroma quality [6,21].

Fatty acids (FAs) are essential constituents of each plant [22,23,24] and various analytical methods for the determination of fatty acids from oil, biological fluids, microalgae, etc. are reported [25]. Lipids are regarded as one of the essential nutrients for humans. Based on the presence of double bonds, lipids are classified as unsaturated fatty acids (SFAs), monounsaturated fatty acids (MUFAs with one double bond), and polyunsaturated fatty acids (PUFAs with two or up to six double bonds). Based on the number of carbon atoms, there are FAs with 14–20 carbon atoms and polyunsaturated fatty acids (PUFAs) with more than 20 carbon atoms in structure (like docosahexaenoic acid (DHA) and eicosapentaenoic acid (EPA)). A high PUFA content is essential for food nutrition [26,27,28,29]. The important PUFAs are α-linolenic acid (C18:3, (n3)) and linoleic acid (C18:2, (n6)). Omega-3 FA was recommended for consumption due to its high properties to help immunity, the brain, and human health [25]. Omega-6 and omega-3 are considered essential FAs because they cannot be synthesized by humans and must be supplemented by food. Moreover, a lack of omega-3 FAs can cause stroke, age-related cognitive decline, and Alzheimer’s disease [25]. Fatty acid types give the aroma of tea and differ depending on the type, maturity, harvest time, and storage time [30]. The efficiency of lipid extraction is dependent on the utilized method, solvent, and type. Usually, classical extraction with solvents is used. Different studies showed that using the pre-treatment method for lipid extraction substantially improved the oil yield. The pre-treatment methods, such as ultrasonication, microwave irradiation, autoclaving, etc., are reported in the literature [31]. A high content of linoleic (C18:2) and palmitic (C16:0) acids and a low content of oleic acid (C18:1) were reported in tea seed oil from Camellia species from the China region [32], and a high content of PUFA (42.79%), followed by SFA (29.18%) and MUFA (12.57%), related to the fatty acid content, were reported for the lavender extract [33]. Moreover, 10% palmitic acid, 2.6% stearic acid, 17% oleic acid, 42% linoleic acid, and 26% linolenic acid were also reported by Dai et al. (2021) for blueberry fruit [34].

Tea comprises mainly polyphenols, polysaccharides, multivitamins, beta-carotene, caffeine, flavonoids, pyrroloquinoline quinone, protein, and amino acids [35]. Tea polyphenols play an important role in fixing metals inside plant cells, protecting the cell constituents against oxidative damage, and consequently limiting the risk of several degenerative diseases related to oxidative stress [36,37]. Moreover, the tea leaves are rich in mineral elements such as Co, Mn, Fe, Zn, Mg, Ca, Na, and K, and they have an important role in the development of tea trees but are also a key expression of the nutritional value of tea since minerals are present in all body tissues and are involved in many life processes [38]. Consequently, frequent consumption of tea may contribute significantly to the recommended daily intake. Ca^2+^ and Fe^3+^ notably exert an influence on the content of alcohols and aldehydes in volatile compounds [39]. Some mineral substances, such as Fe, Zn, Cu, and Se, add value to the human body. Some biological functions they have an effect on include antioxidation, immunomodulation, inhibition of cancer, and playing an important role in energy metabolism and gene expression [39]. On the contrary, toxic mineral substances such as As, Cd, Cr, Ni, and Pb can cause neurotoxicity, nephrotoxicity, and a variety of other adverse effects on health [39]. Mineral-rich water not only deteriorates the flavor profiles but also reduces the contents of catechins and the antioxidant capacity of tea infusions [5]. The factors responsible for the mineral content in plant samples refer to the geographic origin, soil type, use of fertilizers and pesticides, climatic factors, harvest time, type of processing, mining activities, and vehicular emissions [1]. Green tea leaves are rich in bioactive compounds, particularly phenolic compounds exhibiting antioxidant activity, probably due to the low oxidation degree of the young leaves. Polyphenols, which account for 20–30% of the dry weight of tea leaves, are the most abundant class of soluble components that influence the color, taste, and aroma of tea leaves and are the most important substance to exert their health benefits. Tea polyphenolic compounds play an important role in metal binding within plant cells and protect cells from oxidative stress due to their antioxidant properties [35].

Many studies reported the chemical analysis of single or different tea plants, but a complete chemical analysis and correlation between all components still require attention. In this regard, concerning their benefits for human health, the present study aims to compare three tea types (flowers, leaves, and instant) in terms of physicochemical analysis and their thermal behavior.

## 2. Materials and Methods

### 2.1. Chemicals and Reagents

All chemicals and reagents were of analytical grade and were used as received from Merck, except for sodium chlorite purchased from Alfa Aesar. Ultrapure water from a Buckinghamshire Purelab Flex 3 system was used to prepare the standard solutions and dilute the samples.

### 2.2. Tea Samples

Lavender (T1), chamomile (T2), blueberry (T3), Breackland thyme (T4), Yarrow (T5), mint (T6), black (T7), acacia (T8), elderflower (T9), Saint John’s Wort (T10), linden (T11), rose (T12), green (T13), lemon (T14), and pomegranate (T15) were used for this study. Eleven tea samples as mature flowers (T1, T2, T4, T5, T8, T9, T10, T11, and T12) and two mature leaves (T3 and T6) were wild-harvested from an unpolluted area located in Northern Romania. The collected samples were washed, air dried, and ground to a fine powder. Alongside, black leaves (T7), green leaves (T13), instant lemon (T14), and instant pomegranate (T15) tea samples were purchased from a local market and covered different countries of origin. All samples were stored at −20 °C until further processing.

### 2.3. Thermal Analysis

The thermal behavior of the grounded tea sample was investigated by thermogravimetry (TG) and differential thermal analysis (DTA) using a SDTQ600 instrument (TA Instruments, New Castle, DE, USA) in air and argon atmospheres up to 1000 °C at a 10 °C/min heating rate using alumina standards.

### 2.4. Oil Extraction from Teas

The method used to extract oil from tea samples was realized according to a previously published method, with improvements according to Wang et al. (2021) [40]. Approximately 0.5 g were extracted from each sample with 25 mL of chloroform:methanol (2:1, *v*/*v*) in an ultrasonic bath for one hour at room temperature. The organic phase was separated by filtration. Ten milliliters of KCl (0.74%) were added to the organic phase and separated again into a 250 mL separating funnel. The water was removed by filtration using Na_2_SO_4_. A rotary evaporator, Laborota 4010 (Heidolph, Schwabach, Germany), was used to separate the solvent. The obtained oil was weighted and analyzed for fatty acid constituents. The fatty acids were transformed into FAMEs by transesterification with alcohol. The samples (30 mg) were dissolved in 4 mL of isooctane and reacted with 200 µL of methanol potassium hydroxide (CH_5_KO_2_) (2 mol/L) under stirring. One gram of sodium hydrogen sulfate (NaHSO_4_·H_2_O) was added to the final solution. The FAME standard mixture (CRM47885) was acquired from Sigma-Aldrich. The FAMEs content was determined by GC-FID (Agilent Technologies, Santa Clara, CA, USA, 6890N) equipped with a ZB-WAX capillary column (30 m × 0.25 mm × 0.25 µm). Helium was used as the gas carrier at a constant flow rate of 1 mL min^−1^. The furnace temperature started at 60 °C (held for 1 min) and was increased to 200 °C (held for a 2 min plateau) with a rate of 10 °C min^−1^, and, finally, increased to 220 °C (held for a 20 min plateau) with a rate of 5 °C min^−1^. The injector and temperature detector were established at 250 °C. The standard mixture calibration curve was used to quantify all FAs.

### 2.5. Volatile Composition

The volatile compositions of tea samples were determined by headspace-solid phase microextraction (HS-SPME) according to our previously published method [41].

### 2.6. Mineral Profile

The major and trace elements were determined using an inductively coupled plasma optical emission Perkin Elmer Optima 5300DV (ICP-OES) spectrometer after microwave-assisted digestion using a Berghof Xpert system. An amount of 500 mg of ground tea was digested using 4 mL of HNO_3_ at 65% and 6 mL of H_2_O_2_ at 30% in polytetrafluoroethylene digestion vessels using a four-step digestion program (145, 170, and 190 °C—heating; 50 °C—cooling) for a total digestion time of 40 min. Subsequently, the vessels were cooled down, and the volume was made up to the mark with ultrapure water. Blanks were prepared for each lot of samples. The calibration standards were prepared from Merck ICP multi-element standard solution IV, 1000 mg/L (Na, K, Ca, Mg, Fe, Cu, Mn, and Zn), and a mono-element standard solution, 1000 mg/L P. The accuracy in determining the major and trace element concentrations in tea samples was evaluated via NIST SRM 1515 Apple Leaves, achieving satisfactory recoveries (%) of K, Ca, Mg, Fe, Cu, Zn, and P (85.6–105.3%). The total nitrogen (N) was determined by combustion using a Flash EA 2000 CHNS/O analyzer (Thermo Fisher Scientific, Waltham, MA, USA).

### 2.7. Cellulose, Hemicelluloses, and Lignin Content

The amount of cellulose, hemicelluloses, and lignin was established from each tea sample according to our previously published method [41]. The content of holocelluloses (total polysaccharide fraction) was determined as a residue insoluble in sodium chlorite and acetic acid (1%) at 75 °C for 1 h (process repeated four times). The cellulose content was determined as a residue obtained after the reaction of holocellulose with the holocellulose residue insoluble in NaOH (17.5%). The difference between holocellulose and cellulose content was used to determine hemicellulose. The lignin content was established as the insoluble residue in 72% H_2_SO_4_ for 4 h at 20 °C. [41].

### 2.8. Polyphenol Content

The content of polyphenols was determined by the Folin-Ciocalteu colorimetric method using a Lambda 25 spectrophotometer (Perkin-Elmer, Waltham, MA, USA), according to [42].

### 2.9. Statistical Analysis

For all analyses, the mean and standard deviation (SD) were determined for each variable. The multivariate statistical method of hierarchical cluster analysis was employed to find the correlation of the variables and presented as a dendrogram realized with the Minitab Statistics Software (version 21.1. 0, Minitab Inc., PA, USA). The statistical differences of the analyzed parameters were evaluated through Tukey’s test (*p* = 0.05) with the Paired Comparison App (Two-Way ANOVA) by using Origin software (version 2020b, OriginLab, Northampton, MA, USA). The letters indicate a statistically significant difference at *p* < 0.05.

## 3. Results and Discussions

### 3.1. Thermal Behavior

The decomposition stages of the tea samples were investigated by TG-DTA up to 600 °C (Figure 1). On the DTA curve, the tea samples generally show three exothermic effects. However, some samples display six exothermic effects. 

The first stage was set apart by an exothermic effect at around 39–75 °C, accompanied by a mass loss of 1.7–13.5%, which can be attributed to the evaporation of adsorbed water and solvent [7]. The second stage involved the decomposition of volatile compounds, such as polyphenols and fatty acids. It is characterized by (i) a single exothermic effect at 267–319 °C in the case of T1, T2, T3, T4, T5, T6, T7, T10, and T11 samples, accompanied by a mass loss of 37.2–50.3%; (ii) two exothermic effects at 139–185 °C/298–315 °C in the case of T9, T13, and T15 samples, accompanied by a mass loss of 43.1–59.6%; and (iii) three exothermic effects at 143–188 °C/211–217 °C/304–329 °C in the case of T8, T12, and T14 samples, accompanied by a mass loss of 43.1–59.6% [7]. The third stage of decomposition, corresponding to the degradation of cellulose, hemicellulose, and lignin, was generally noticeable through a single exothermic effect at 424–504 °C, followed by a mass loss of 32.5–52.6%. In the case of T2 and T3 samples, two close effects on the same intense peak at 424–444 °C/438–480 °C with a mass loss of 32.5–40.8% and three exothermic effects at 432 °C/477 °C/496 °C with a mass loss of 34.1% in the case of T13 sample (green tea), corresponding to the decomposition of hemicellulose, cellulose, and lignin, were noted [7,43]. The total mass losses were in the range of 94.4–100%. Hemicellulose, a compound that possesses a lower degree of polymerization, is a mixture of various polymerized monosaccharides (xylose, mannose, glucose, galactose, and arabinose). At the same time, cellulose is a high-molecular-weight compound consisting of a long linear chain of D-glucosyl groups. The crystalline structure of cellulose renders its thermal degradation more difficult than that of hemicellulose. In addition, on account of the complex composition, the degradation of lignin and macromolecular substances was also more strenuous than that of hemicellulose [43].

### 3.2. Cellulose, Hemicelluloses, and Lignin Content of Tea Samples

In all tea samples, cellulose, hemicelluloses, and lignin were detected. The cellulose, hemicelluloses, and lignin contents were 5.6–19.3%, 5.2–20.4%, and 11.9–26.8%, respectively. The cellulose (11.6%), hemicelluloses (20.4%), and lignin (20.8%) content determined for T6 were in line with the results obtained by other researchers. Furthermore, 7.06% cellulose, 20.64% hemicelluloses, and 20.04% lignin were also published by Cai et al. (2019) for mint tea [43]. The compositions of cellulose, hemicelluloses, and lignin vary depending on tea plant constituents (flowers, leaves, and branches). Lesage-Meesen et al. (2015) reported 22.4% cellulose, 12.6% hemicelluloses, and 23.6% lignin from the lavender flower; 19.9% cellulose, 3.6% hemicelluloses, and 17.1% lignin from lavender leaves; and 42.7% cellulose, 13.0% hemicelluloses, and 23.1% of lavender branches [44]. Generally, the cellulose, hemicellulose, and lignin content differs by function, season, plant component (flowers, branches, and leaves), and climatologic conditions. In addition, the main components of tea give it rigidity and strength. The highest lignin concentration was found in black tea leaves (26.8%), due to their high syringyl content. The content of cellulose varied in the following order: T1 (19.3%) > T3 (18.5%) > T13 (18.1%) > T7 (16.4%) > T5 (14.2%) > T9 (13.2%) > T11 (12.8%) > T6 (11.6%) >T4 (10.7%), T10 (10.5%), T8 (10.2%), T12 (10.1%). Lignin components are the essential compounds that contribute to plant growth, structure integrity, and water transport. Lignin is the second component in plants after cellulose and contains monolignol (coumaryl, coniferyl, and sinapyl alcohol) and lignan (dimers of monolignols) [45]. The content of lignin is generally higher in leaves than in flowers. The higher lignin content was found in T11 (26.9%), T7 (26.8%), and T1 (24.5%). The hemicellulose content is higher in T6 (20.4%) and T7 (18.6%) and smaller in T9 (5.2%). The presence of cellulose, hemicelluloses, and lignin was confirmed by thermogravimetric analysis. The content of lignin varied in the following order: T11 (26.9%) > T7 (26.8%) > T13 (24.5%) > T1 (22.6%) > T8 > T6 (20.8%) >T3 (20.68%) >T4 (20.3%) > T10 (17.9%) > T9 (17.2%) > T5 (15.9%) > T2 (15.6%) > T12 (11.9%). In the instant tea samples, T14 and T15, cellulose, hemicelluloses, and lignin content were not identified.

### 3.3. HS-SPME GC-MS Analysis of Volatile Organic Compounds

Headspace-solid phase microextraction (HS-SPME) is regularly employed for extracting volatile aroma compounds and consists of isolating volatile substances in the headspace of the sample vial via a polymer fiber, followed by their detection using gas chromatography-mass spectrometry (GC-MS) [6]. The present study determines 66 volatile compounds that can be divided into eight classes within the tea samples, including 29 terpenes, 12 hydrocarbons, 9 esters, 7 aldehydes, 5 ketones, 2 alcohols, 1 furan, and 1 thiofuran (Table 1). The content and proportion of these groups varied in the studied tea samples; they were either diminished or newly generated, possibly due to the release of volatile compounds from the Maillard reaction, carotenoids or lipid degradation, and glycoside hydrolysis during the tea manufacturing process [2]. The drying method and temperature had a great effect on the relative content of volatile components and influenced the formation of aroma types [46]. The aroma of tea can be influenced by the plant cultivar, environment, and cultivation, as well as the processing technology [46].

Terpenes display the highest proportion (73.0%) in tea samples, inducing fruity, sweet, lemony, pine-like, minty, woody, balsamic, citrusy, floral, aromatic, green-like, or herbaceous [47]. Terpenoids, and mainly the C_10_ and C_15_ members within this family, generally impact the flavor profiles of most fruits as well as the scent of flowers. Moreover, the synthetic variations and derivatives of natural terpenes and terpenoids can expand upon the aroma varieties used in perfumery and as flavorings [48]. Limonene was predominant in all tea samples, although its amount varied among them as follows: T15 (97.8%) > T14 (61.2%) > T13 (23.8%). Limonene has ‘lemon’, ‘sweet orange peel’, and ‘licorice’ odors, which play an essential role in the unique aroma of tea [49]. A higher m-cymene content was detected in sample T7. Eucalyptol was the principal group of volatiles identified within the tea samples, although the amounts varied between samples as follows: T8 (62.8%) > T1 (43.2%) > T6 (27.6%) > T13 (17.3%). Eucalyptol, mainly found in sample T8, is utilized as a flavoring and fragrance ingredient due to its fresh and pleasant smell, spicy taste, and cooling effect [50]. Linalool is a monoterpene alcohol that accounts for around 70% of floral fragrances and is employed as a flavoring and air scent aside from its antibacterial and insecticidal properties [51]. 4-Terpineol displaying ‘pepper’, ‘woody’, and ‘earthy’ odors was also detected as one of the odor-active compounds in tea samples [40]. Linalool contributes to the sweet, tender, and fresh floral aroma with a specific smell of bergamot, which was the main contribution to the aroma characteristics of tea [52]. Carvone provided ‘mint’, ‘basil’, and ‘fennel’ odors, which were detected as one of the odor-active compounds of tea [49].

Hydrocarbons (10.6%) in tea samples induce fruity, camphoraceous, sweet, lemony, pine-like, minty, woody, resinous, balsamic, plastic, roasted, citrusy, floral, aromatic, green, chemical, herbaceous, clove, pepper, or spicy odors [47]. The literature studies should demonstrate that tea leaves accumulate aromatic hydrocarbons through airborne deposition in the environment and during treatment stages, such as drying by burning wood or coal [53].

Esters (6.0%) present in tea samples induced fruity (apple, pear, or cherry, among others), floral, herbaceous, sweet, refreshing, green, grassy, bergamot, lavender, and minty odors. Among these, ethyl salicylate can be described as having notes of ‘wintergreen’ and ‘mint’, and it was revealed as being an odor-active compound in the T9 sample [49].

Aldehydes (5.2%) are obtained through the oxidative degradation of amino acids during their interaction with sugars at high temperatures or through the interaction of amino acids and polyphenols in the presence of polyphenol oxidase [2,54]. Benzaldehyde, the singular odor-active aromatic aldehyde with the highest content in sample T4, is characterized by a caramel-like or roasted odor and commonly exists as the glycosidically bound form in tea [2]. The aromas conferred by aldehydes are green, fruity, fatty, grassy, oily, very strong, harsh, and lemony [52]. 2-Hexenal, providing ‘apple’ and ‘green’ odors, contributed to rich, fresh fruit and green leaf fragrance; n-heptanal provided ‘pungent’ and ‘unpleasant’ odors [49,52].

In all tea samples, ketones (4.6%) are associated with fruity, spicy, cinnamon, banana, mushroom, camphor, cedar leaf, mint, and bitter aromas [47]. Ketones are commonly derived from the degradation of carotenoids/unsaturated fatty acids and the hydrolysis of their glycoside precursors [54]. 7-Octen-2-one, described as possessing ‘fatty’, ‘fruity’, and ‘mushroom’ odors, was identified as the primary aroma compound degraded during the post-fermentation of tea [49]. 2-Heptanone contributed to ‘stale’ and ‘cabbage’ odors, but little contributed to the entire aroma formation of tea because of its high threshold and low presence [49].

The lowest content of alcohol (0.2%) accounts for cooling, camphoraceous, fresh pine, ozone, citrus, floral, green, peppermint, woody, earthy, and sweet odors [2].

Various compounds with higher content in different teas were described as having fruity or floral aromas, according to Fenaroli’s Handbook of Flavor Ingredients [47]. All these volatile compounds are classified into nine odor classes (floral, fruity, green, sweet, chemical, woody, citrus, roasted, and alcohol) (Table 1, Figure 2). The most volatile compounds are present in St. John’s wort tea (T10). Generally, fruity (35.1%) and floral (23.9%) aromas prevailed in the studied samples. The predominant fruity aroma was found in T1, T2, T3, T4, T5, T6, T8 (the most intense), T11, and T12. The predominant floral aroma was noted in T6 (the most intense), T7, and T10. The green aroma prevailed in the T sample, while the citrus aroma predominated in the T15, T14, and T13 samples. The flower aroma of α-pinene was found in all studied tea samples except for the T15 sample. Linalool is a kind of monoterpene alcohol that is rich in lavender, St. John’s wort, rose, thyme, chamomile, elder, linden, acacia, and blueberry teas and is associated with floral, citrus, and fruit flavors [4].

### 3.4. FAMEs Content of Tea Samples

The FA contents of all tea-analyzed oils are presented in Table 2. SFAs, MUFAs, and PUFAs were identified in all oil tea extracts. The SFA classes include caprylic acid, capric acid, lauric acid, myristic acid, palmitic acid, stearic acid, arachidic acid, eicosadienoic acid, tricosanoic acid, and lignoceric acid. The main SFAs present in all tea samples are palmitic acid (C16:0), followed by arachidic acid (C20:0) and stearic acid (C18:0). The MUFA included nine fatty acids: C14:1 (n-9), C15:1, C16:1 (n-7), C17:1, C18:1 (cis + trans) (n-9), C20:1 (n-9), C22:1 (n-9), and C24:1 (n-9). The peaks that overlap are isomers of oleic and linoleic acids. Moreover, *cis*-8,11,14-eicosatrienoic acid coelutes with heneicosanoic acid, and cis-4,7,10,13,16,19-docosa-hexanoic acid coelutes with nervonic acid.

The SFAs, MUFAs, PUFAs, and omega-3 and omega-6 content in the tea oil are presented in Figure 3. The MUFA component found to be most abundant is oleic acid, and its content varied in the following order: T3 > T9, T2 > T8 > T1 > T4 > T11 > T7 > T5 > T10 > T6 > T6 > T13 > T12. The greatest content of linoleic acid C18:2 (cis + trans) (n-9) was found in T5 (11.5%), followed by T8 (10.14%) and T2 (10.07%). The consumption of foods with a high content of PUFA has a positive influence on cardiovascular disease. The acacia tea samples contain the highest content of omega-6 (53.0%), while the lowest amount was identified in the Saint John’s Wort sample (11.0%). Acacia, Saint John’s Wort, rose, and yarrow are good sources of omega-6. The consumption of 3 g of omega-3 can reduce hypertension problems among all the acids [55]. Good sources of omega-3s are tea samples T6, T10, T2, T13, T3, and T1.

The content of fatty acids reported in peppermint (T6) is in good agreement with the obtained results of Maffei et al. (1992), e.g., the FA fatty acids dominate in palmitate (16:0), linoleate (18:2), and linolenate (18:3) [56]. Eicosapentaenoic acid (EPA, C20:5, n-3) was found in small quantities in T5 and T10.

A high quantity of C20:2 (n-6) (PUFA) was found in T12 (24.04%). The 22:6 (n-3) PUFA was found only in thyme (T4) (6.5%) and T2 oils (6.0%). Moreover, thyme oil contains 20:4 (n-6) PUFA. The highest PUFA content was found in acacia (T8) (53%). Myristoleic acid (C14:1) was not identified in samples of acacia and linden oil.

### 3.5. Mineral Compositions and the Content of Total Polyphenols

The studied tea samples are abundant in major elements (Na, K, Ca, Mg, and P), which play major roles in both the growth and development of tea trees and the nutritional value of tea. The content of major and trace elements and polyphenols in the tea samples is highlighted in Table 3. Appreciable amounts of Na, K, Ca, Mg, and P were observed in all samples, except in the instant tea samples (T14 and T15). The major element content was reduced in the following order: K > Ca > Mg > P > Na, while the trace element content decreased in the Fe > Zn > Cu sequence. Excepting the instant tea samples, the lowest mineral and nitrogen content was determined for the T1 (lavender-flower) sample. A possible explanation for the chemical composition of different tea types could be reflected in the differences in varieties, growing soil, and manufacturing techniques that different tea leaves have been exposed to during their different phases of growth and processing [57]. Minerals play key roles in the body, from building strong bones to transmitting nerve impulses for a healthy and lengthy life [58]. Na, K, and Mg are involved in the body’s control, treatment, and management of various metabolic and heart disorders, while Ca is associated with healthy bones and teeth, although it plays an important role in maintaining extracellular fluids, blood clotting, and muscle contraction [59]. Phosphorus has a critical role in metabolism and growth, whereas Fe is a crucial element in living systems, being involved in the biosynthesis of hemoglobin in erythrocytes and the physiology of oxygen transportation [58]. In addition, nitrogen is one of the most beneficial nutritional elements for the tea plant, having a high impact on the overall quality of tea. The trace elements present within tea leaves form complexes with flavonols, tannins, catechols, and polyphenols [57]. Polyphenols present in tea are an order of phytochemicals possessing multiple phenolic groups, some of which are flavonoids and their glucosides, phenolic acids, and any other compounds containing phenolic hydroxyl groups (such as tyrosine). They are connected to health-promoting properties such as antidiabetic, antioxidant, anti-inflammatory, and anticarcinogenic. Catechins are revealed to reach a level of about 60–70% of tea polyphenols within the fresh tea leaves [20]. 

### 3.6. Multivariate Analysis

Hierarchical clustering (dendrogram) of the HCA of the tea samples is highlighted in Figure 4 and was realized by considering all the variables used for tea characterization. According to cluster analysis, two main clusters can be observed. The first group contains T1, T10, T13, T2, T5, T6, T7, T8, T11, and T9, and the second group contains T3, T4, T12, T14, and T15. The first cluster was further separated into three subclusters, and the second cluster was divided into two subclusters. The chemical composition of lavender is similar to that of Saint John’s Wort and green leaves, and the chemical composition is similar for chamomile, Yarrow, mint, and black.

The HCA group of tea samples based on a set of variables like metals, polyphenols, carbohydrates (cellulose and hemicelluloses), and lignin is presented in Figure 5. Two clusters were formed, the first cluster containing Na, K, P, N, Cu, Mg, and Zn, and the second cluster containing Ca, Fe, polyphenols, hemicellulose, cellulose, and lignin. Hemicelluloses are cell wall polysaccharide fractions from tea samples and contain heteropolysaccharides, and polyphenols can be bound by polysaccharides by an adsorption system [60]. According to Liu (2017), the association between polyphenols and cellulose was possible due to hydrogen bonding linkage formation [61]. The polyphenols can bind to the cellulose surface; this process was accelerated in water. In addition, the metal content, particularly Ca, contributes to the growth and development of the tea plant, and Fe is required in a small amount and is used by the plant as a nutrient and for reproduction.

The other macronutrients, like nitrogen, potassium, and phosphorus, contribute to the tea plant’s growth. The HCA of different tea samples (Figure 6), considering the fatty acids and volatile compounds, obtained the following correlations: T6 correlated with T13 due to the high C18:3 (n-3) found in both samples and the correlation between T4, T11, T12, T6, and T13 due to abundant hydrocarbons correlated with PUFA.

## 4. Conclusions

The present work examines a comparative analysis of fatty acids, volatile compounds, and minerals and presents an evaluation of the thermal behavior among different types of tea (leaves, flowers, and instant) samples. The volatile compound and aroma description profiles were also identified. The decomposition of the studied tea samples took place in three stages: moisture evaporation at 39–75 °C, followed by a mass loss of 1.7–13.5%; decomposition of volatile compounds and fatty acids at 139–329 °C, characterized by 1–3 exothermic effects and attended by a mass loss of 37.2–57%; and degradation of cellulose, hemicellulose, and lignin at 424–504 °C, characterized by 1–3 exothermic effects and accompanied by a mass loss of 32.5–52.6%. The total mass losses were in the range of 94.4–100%. The predominant aromas were fruity in T1, T2, T3, T4, T5, T6, T8 (most intense), T11, and T12 samples; floral in T6 (most intense), T7, and T10 samples; and green in the T9 sample, while the citrus aroma predominated in the T15, T14, and T13 samples. A total of 66 volatile compounds were separated into eight classes within the tea samples. This includes 29 terpenes, 12 hydrocarbons, 9 esters, 7 aldehydes, 5 ketones, 2 alcohols, 1 furan, and 1 thiofuran. The major element content decreased in the order (mg/kg): K (29.3–30,400) > Ca (1251–19,504) > Mg (11.9–3345) > P (286–5545) > Na (24.8–669), while the trace element content decreased in the order Fe (10.9–3912) > Zn (4.30–41.7) > Cu (1.10–7.70). In all flower and leaf teas, SFAs, MUFAs, and PUFAs were identified. A high content of omega-6 was quantified in acacia, Saint John’s Wort, rose, and yarrow, while omega-3 was found in mint, Saint John’s Wort, green, blueberry, and lavender samples. Some minerals (Fe, Cu, and Zn) were not identified in instant tea samples, and a low quantity of polyphenols and volatile compounds was obtained. Overall, it can be said that instant tea has fewer nutritional properties than brewed tea.

## Figures and Tables

**Figure 1 foods-12-03063-f001:**
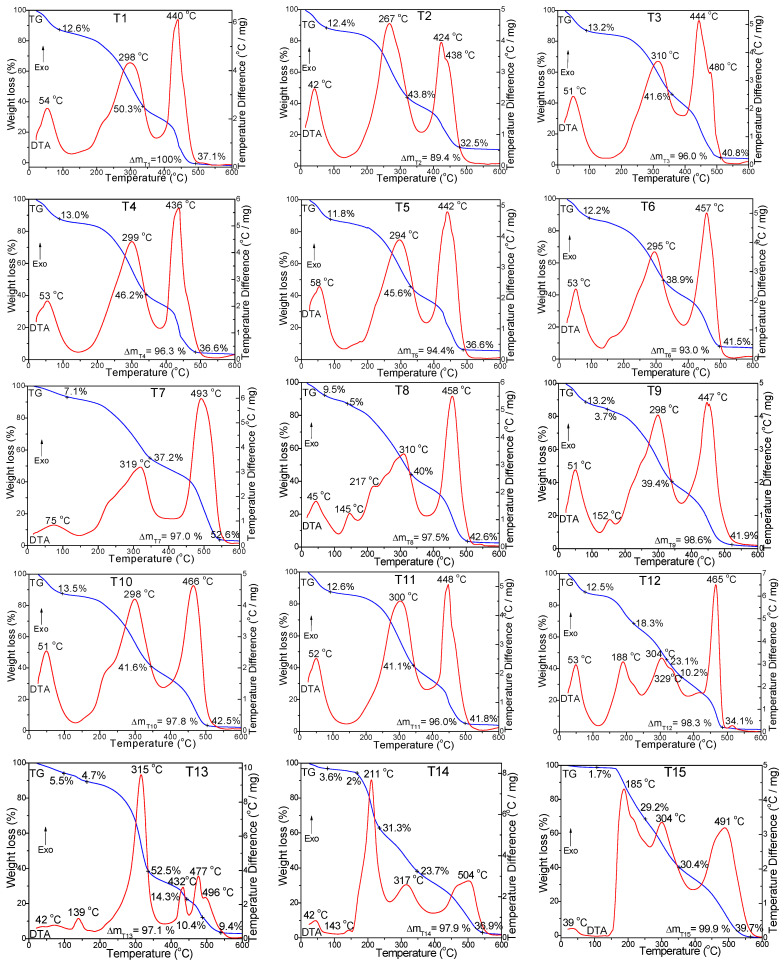
TG (blue line) and DTA (red line) curves of the tea sample.

**Figure 2 foods-12-03063-f002:**
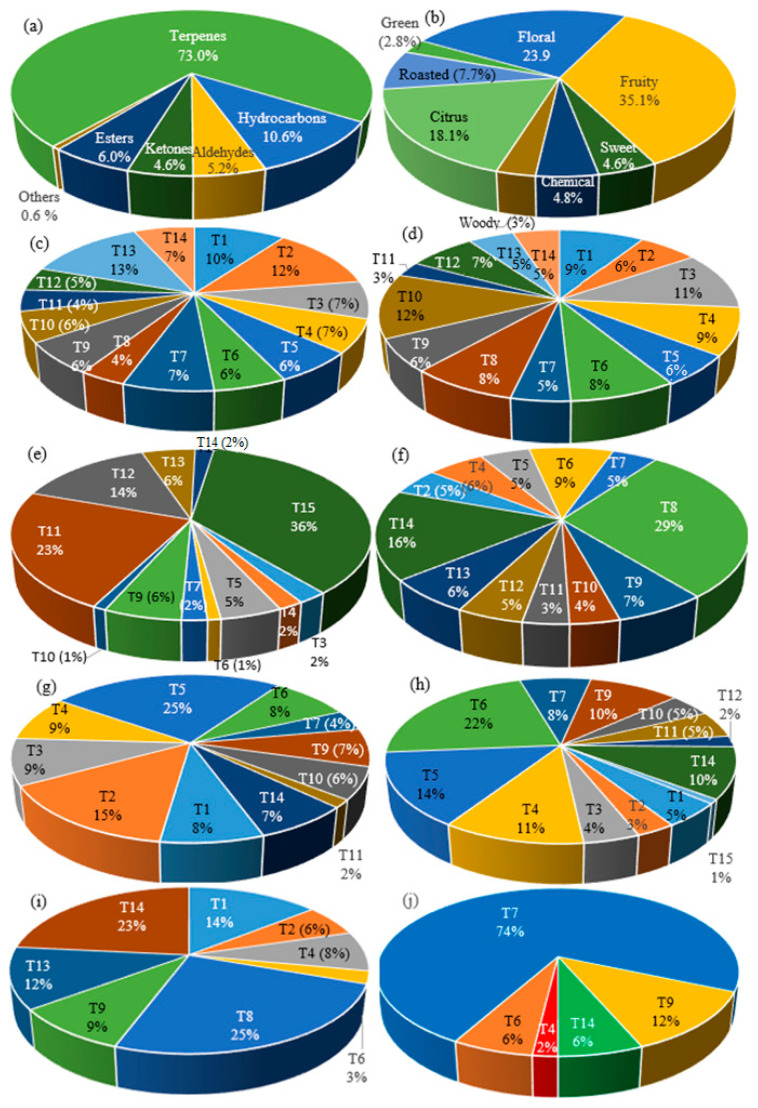
Classification of volatile compounds identified in teas by chemical class (**a**) and aroma profile (**b**): floral (**c**), fruity (**d**), citrus (**e**), roasted (**f**), Chemical (**g**), sweet (**h**), woody (**i**), and green (**j**) aroma tea distribution.

**Figure 3 foods-12-03063-f003:**
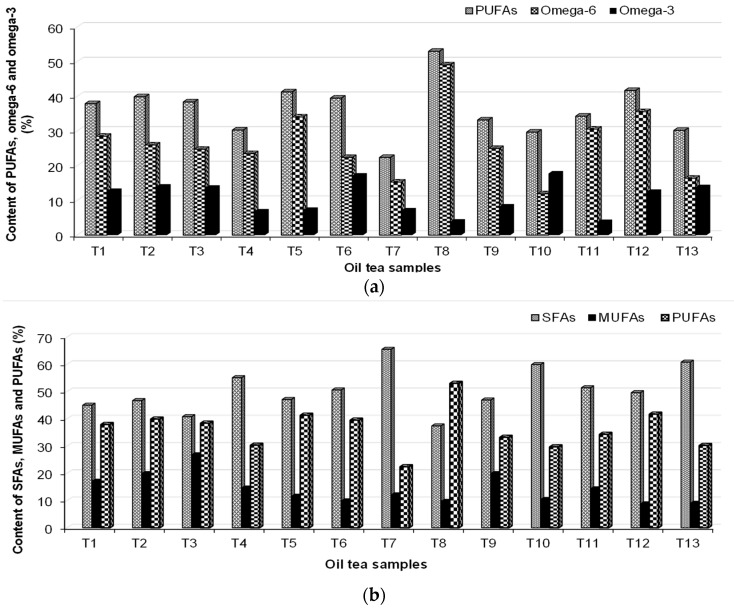
The content of (**a**) SFAs, MUFAs, and PUFAs and (**b**) PUFAs, omega-6, and omega-3 in the analyzed oil tea samples.

**Figure 4 foods-12-03063-f004:**
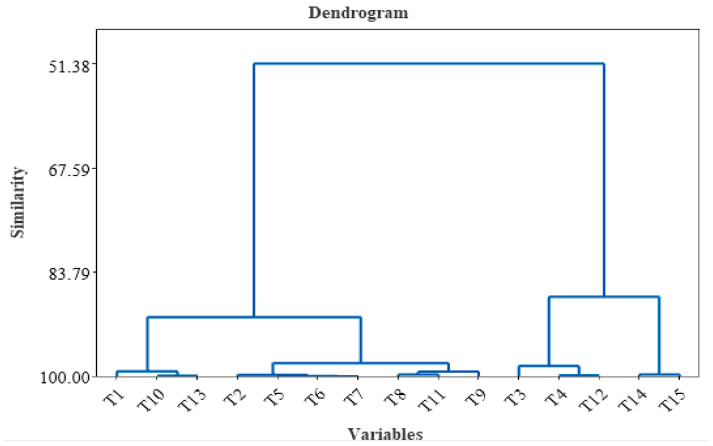
Hierarchical clustering (dendrogram) of different tea samples T1–T15 (including all variable analysis).

**Figure 5 foods-12-03063-f005:**
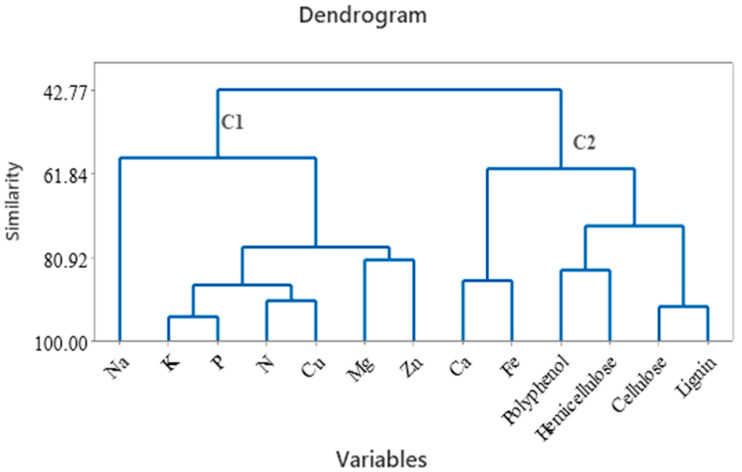
Hierarchical clustering (dendrogram) of metals, cellulose, hemicellulose, lignin, and polyphenols from tea samples.

**Figure 6 foods-12-03063-f006:**
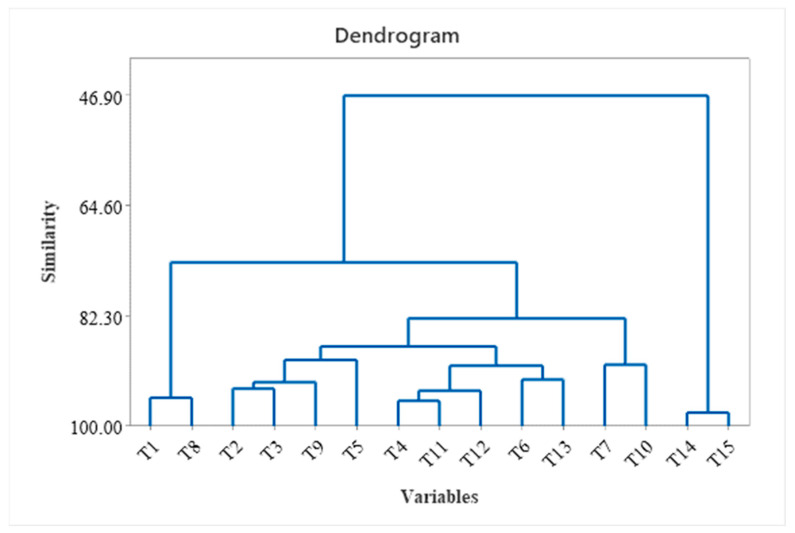
Hierarchical clustering (dendrogram) of different tea samples T1–T15 (fatty acids and volatile compounds).

**Table 1 foods-12-03063-t001:** Retention time, volatile organic compounds, formula, group, odor type, and content (%) identified by HS-SPME GC-MS for T1–T15 tea samples. Data are expressed as a mean ± standard deviation (n = 3).

R_t_ (min)	Volatile Compounds	Molecular Formula	Group	Odour Types	T1	T2	T3	T4	T5	T6	T7	T8	T9	T10	T11	T12	T13	T14	T15
5.1	2-methylbutan-1-ol	C_5_H_12_O	Alcohols	green	<0.03	0.9 ± 0.1	<0.03	<0.03	<0.03	<0.03	<0.03	<0.03	<0.03	<0.03	<0.03	<0.03	<0.03	<0.03	<0.03
6.4	isobutyl acetate	C_6_H_12_O_2_	Esters	floral	<0.02	<0.02	<0.02	<0.02	<0.02	<0.02	<0.02	<0.02	6.6 ± 0.7	<0.02	<0.02	<0.02	<0.02	<0.02	<0.02
6.5	methyl isovalerate	C_6_H_12_O_2_	Esters	fruity	<0.04	1.6 ± 0.1	<0.04	<0.04	<0.04	<0.04	<0.04	<0.04	<0.04	<0.04	<0.04	<0.04	<0.04	<0.04	<0.04
7.0	pentanal	C_5_H_10_O	Aldehydes	roasted	<0.02	<0.02	2.7 ± 0.3 ^a^	<0.02	<0.02	<0.02	<0.02	<0.02	<0.02	<0.02	1.5 ± 0.1 ^b^	<0.02	<0.02	<0.02	<0.02
7.4	hexanal	C_6_H_12_O	Aldehydes	green	<0.01	<0.01	2.7 ± 0.3 ^c^	<0.01	<0.01	<0.01	<0.01	<0.01	8.2 ± 0.9 ^a^	<0.01	5.1 ± 0.6 ^b^	<0.01	<0.01	<0.01	<0.01
9.2	3-benzyloxypropan-1-ol	C_10_H_14_O_2_	Alcohols	alcohol	<0.02	<0.02	<0.02	<0.02	<0.02	<0.02	<0.02	<0.02	1.2 ± 0.1	<0.02	<0.02	<0.02	<0.02	<0.02	<0.02
9.6	2-hexenal	C_6_H_10_O	Aldehydes	green	<0.02	0.4 ± 0.01 ^b^	<0.02	<0.02	<0.02	<0.02	<0.02	<0.02	9.9 ± 1.1 ^a^	<0.02	<0.02	<0.02	<0.02	<0.02	<0.02
9.6	2-methyl-4-pentanal	C_6_H_10_O	Aldehydes	green	<0.03	1.3 ± 0.08 ^a^	<0.03	<0.03	<0.03	<0.03	<0.03	<0.03	<0.03	<0.03	<0.03	0.8 ± 0.08 ^b^	<0.03	<0.03	<0.03
10.1	2-methyloctane	C_9_H_20_	Hydrocarbons	chemical	<0.01	<0.01	<0.01	<0.01	<0.01	0.8 ± 0.02 ^b^	<0.01	<0.01	<0.01	7.6 ± 0.8 ^a^	<0.01	1.1 ± 0.1 ^b^	<0.01	<0.01	<0.01
11.1	styrene	C_8_H_8_	Hydrocarbons	floral	<0.04	<0.04	3.9 ± 0.4 ^b^	<0.04	<0.04	<0.04	<0.04	<0.04	<0.04	<0.04	12.1 ± 1.3 ^a^	<0.04	<0.04	<0.04	<0.04
11.2	2-heptanone	C_7_H_14_O	Ketones	fruity	<0.03	<0.03	5.9 ± 0.6 ^a^	<0.03	<0.03	<0.03	<0.03	<0.03	3.8 ± 0.4 ^b^	<0.03	<0.03	<0.03	1.2 ± 0.1 ^c^	<0.03	<0.03
11.5	heptanal	C_7_H_14_O	Aldehydes	fruity	<0.05	4.0 ± 0.3 ^b^	<0.05	<0.05	<0.05	2.1 ± 0.3 ^cd^	<0.05	0.6 ± 0.1 ^d^	<0.05	14.1 ± 1.5 ^a^	4.6 ± 0.5 ^b^	3.5 ± 0.4 ^bc^	<0.05	<0.05	<0.05
11.7	nonane	C_9_H_20_	Hydrocarbons	chemical	<0.04	<0.04	1.9 ± 0.01	<0.04	<0.04	<0.04	<0.04	<0.04	<0.04	<0.04	<0.04	<0.04	<0.04	<0.04	<0.04
11.9	santolina triene	C_9_H_14_	Hydrocarbons	fruity	<0.02	0.3 ± 0.01 ^c^	2.0 ± 0.2 ^b^	<0.02	3.6 ± 0.4 ^a^	<0.02	<0.02	<0.02	<0.02	<0.02	<0.02	2.5 ± 0.3 ^b^	<0.02	<0.02	<0.02
12.3	3-carene	C_10_H_16_	Terpenes	sweet	0.9 ± 0.02 ^c^	0.3 ± 0.01 ^d^	<0.03	3.6 ± 0.4 ^a^	<0.03	0.2 ± 0.1 ^d^	<0.03	<0.03	<0.03	<0.03	<0.03	<0.03	1.5 ± 0.1 ^b^	<0.03	<0.03
12.6	4-carene	C₁₀H₁₆	Terpenes	fruity	<0.05	0.4 ± 0.01 ^b^	<0.05	2.6 ± 0.3 ^ab^	<0.05	<0.05	17.8 ± 1.8 ^a^	<0.05	<0.05	0.9 ± 0.01 ^b^	<0.05	4.6 ± 0.5 ^ab^	1.7 ± 0.1 ^b^	<0.05	<0.05
12.8	α-pinene	C_10_H_16_	Terpenes	floral	2.7 ± 0.2 ^ef^	10.2 ± 1.2 ^c^	8.1 ± 0.9 ^cd^	8.6 ± 0.9 ^c^	7.6 ± 0.8 ^cd^	7.1 ± 0.7 ^cd^	7.9 ± 0.8 ^cd^	3.9 ± 0.4 ^e^	3.5 ± 0.3 ^e^	29.3 ± 3.1 ^a^	7.1 ± 0.7 ^cd^	13.4 ± 1.4 ^b^	5.4 ± 0.6 ^de^	0.7 ± 0.02 ^f^	<0.03
13.4	camphene	C_10_H_16_	Terpenes	chemical	5.6 ± 0.1 ^ab^	4.8 ± 0.6 ^ab^	3.3 ± 0.4 ^b^	17.9 ± 1.8 ^a^	<0.03	5.6 ± 0.6 ^ab^	<0.03	4.4 ± 0.5 ^ab^	2.8 ± 0.3 ^b^	0.8 ± 0.01 ^b^	5.2 ± 0.6 ^ab^	5.2 ± 0.6 ^ab^	<0.03	1.1 ± 0.1 ^b^	<0.03
13.8	benzaldehyde	C₇H₆O	Aldehydes	roasted	<0.04	0.6 ± 0.02 ^b^	<0.04	<0.04	<0.04	<0.04	<0.04	1.4 ± 0.2 ^c^	2.9 ± 0.3 ^a^	<0.04	<0.04	3.5 ± 0.4 ^ab^	<0.04	<0.04	<0.04
14.3	3-methylnonane	C_10_H_22_	Hydrocarbons	fruity	<0.02	2.4 ± 0.1 ^c^	<0.02	<0.02	<0.02	1.4 ± 0.2 ^c^	<0.02	<0.02	<0.02	7.1 ± 0.8 ^a^	<0.02	4.2 ± 0.5 ^b^	<0.02	<0.02	<0.02
14.3	β-phellandrene	C_10_H_16_	Terpenes	fruity	<0.02	3.2 ± 0.4 ^de^	<0.02	4.5 ± 0.5 ^d^	18.2 ± 2.1 ^a^	3.8 ± 0.4 ^de^	3.4 ± 0.4 ^de^	<0.02	<0.02	1.7 ± 0.2 ^e^	6.7 ± 0.7 ^c^	10.9 ± 1.2 ^b^	4.7 ± 0.5 ^cd^	<0.02	<0.02
14.4	carene 4.5-epoxy-trans	C_10_H_16_O	Terpenes	sweet	1.1 ± 0.1	<0.04	<0.04	<0.04	<0.04	<0.04	<0.04	<0.04	<0.04	<0.04	<0.04	<0.04	<0.04	<0.04	<0.04
14.4	β-pinene	C_10_H_16_	Terpenes	roasted	<0.02	8.8 ± 0.9 ^c^	16.2 ± 1.7 ^b^	5.9 ± 0.6 ^cd^	33.5 ± 3.5 ^a^	<0.02	7.4 ± 0.8 ^c^	2.8 ± 0.3 ^d^	2.5 ± 0.3 ^d^	5.8 ± 0.6 ^cd^	<0.02	<0.02	5.6 ± 0.6 ^cd^	2.8 ± 0.3 ^d^	<0.02
14.9	7-octen-2-one	C_8_H_14_O	Ketones	fruity	<0.03	0.3 ± 0.01 ^d^	4.8 ± 0.5 ^ab^	3.0 ± 3.3 ^c^	<0.03	5.8 ± 0.6 ^a^	<0.03	<0.03	1.3 ± 0.2 ^d^	<0.03	4.6 ± 0.6 ^b^	5.4 ± 0.6 ^ab^	<0.03	<0.03	<0.03
15.1	β-myrcene	C_10_H_16_	Terpenes	sweet	1.3 ± 0.08 ^f^	14.8 ± 0.2 ^a^	6.9 ± 0.7 ^bc^	6.4 ± 0.7 ^bc^	<0.04	2.8 ± 0.3 ^ef^	<0.04	3.2 ± 0.4 ^ef^	5.4 ± 0.6 ^cd^	2.1 ± 0.2 ^ef^	6.9 ± 0.7 ^bc^	7.6 ± 0.8 ^b^	<0.04	3.6 ± 0.4 ^b^	0.6 ± 0.01 ^de^
15.5	1,3,8-p-menthatriene	C_10_H_14_	Terpenes	roasted	<0.02	<0.02	<0.02	<0.02	<0.02	<0.02	<0.02	<0.02	<0.02	<0.02	<0.02	<0.02	<0.02	1.1 ± 0.2	<0.02
15.7	terpinyl acetate	C_12_H_20_O_2_	Terpenes	green	<0.04	<0.04	<0.04	<0.04	<0.04	<0.04	<0.04	<0.04	13.1 ± 1.5	<0.04	<0.04	<0.04	<0.04	<0.04	<0.04
15.8	n-hexyl acetate	C_8_H_16_O_2_	Esters	fruity	<0.03	<0.03	<0.03	<0.03	<0.03	<0.03	<0.03	<0.03	4.8 ± 0.5	<0.03	<0.03	<0.03	<0.03	<0.03	<0.03
15.9	α-phellandrene	C_10_H_16_	Terpenes	fruity	<0.03	<0.03	<0.03	<0.03	2.6 ± 0.3 ^b^	<0.03	<0.03	<0.03	7.2 ± 0.8 ^a^	0.6 ± 0.02 ^c^	<0.03	<0.03	<0.03	<0.03	<0.03
16.0	p-cymene	C_10_H_14_	Hydrocarbons	citrus	<0.02	<0.02	<0.02	<0.02	<0.02	0.4 ± 0.1 ^b^	<0.02	<0.02	<0.02	<0.02	9.7 ± 1.1 ^a^	<0.02	<0.02	<0.02	<0.02
16.2	m-cymene	C_10_H_14_	Hydrocarbons	floral	<0.03	0.8 ± 0.02 ^d^	6.9 ± 0.8 ^c^	<0.03	<0.03	2.8 ± 0.3 ^d^	33.7 ± 3.5 ^a^	1.8 ± 0.2 ^d^	1.4 ± 0.2 ^d^	2.4 ± 0.3 ^d^	<0.03	3.1 ± 0.4 ^d^	2.4 ± 0.3 ^d^	11.3 ± 1.3 ^b^	<0.03
16.3	limonene	C_10_H_16_	Terpenes	citrus	<0.04	2.7 ± 0.2 ^e^	5.3 ± 0.6 ^de^	5.6 ± 0.6 ^de^	<0.04	2.6 ± 0.3 ^e^	15.1 ± 1.6 ^cd^	2.7 ± 0.3 ^e^	5.2 ± 0.6 ^e^	0.9 ± 0.02 ^e^	6.2 ± 0.7 ^de^	5.1 ± 0.6 ^e^	23.8 ± 2.5 ^c^	61.2 ± 6.5 ^b^	97.8 ± 9.9 ^a^
16.4	eucalyptol	C_10_H_18_O	Terpenes	fruity	43.2 ± 5.1 ^b^	7.1 ± 0.7 ^ef^	11.7 ± 1.3 ^def^	14.5 ± 1.6 ^de^	14.5 ± 1.6 ^de^	27.6 ± 3.1 ^c^	<0.02	62.8 ± 6.6 ^a^	6.3 ± 0.7 ^f^	4.5 ± 0.5 ^f^	16.4 ± 1.8 ^d^	8.7 ± 0.9 ^ef^	17.3 ± 1.9 ^d^	<0.02	<0.02
16.6	decane	C_10_H_22_	Hydrocarbons	floral	0.4 ± 0.01	<0.02	<0.02	<0.02	<0.02	<0.02	<0.02	<0.02	<0.02	<0.02	<0.02	<0.02	<0.02	<0.02	<0.02
17.0	β-ocimene	C_10_H_16_	Terpenes	fruity	0.3 ± 0.01 ^c^	<0.03	<0.03	1.9 ± 0.2 ^a^	<0.03	<0.03	<0.03	0.9 ± 0.02 ^b^	<0.03	<0.03	<0.03	<0.03	<0.03	<0.03	<0.03
17.4	γ-terpinene	C_10_H_16_	Terpenes	fruity	<0.02	<0.02	<0.02	2.1 ± 0.3 ^cd^	3.2 ± 0.4 ^bcd^	<0.02	4.5 ± 0.5 ^b^	<0.02	<0.02	1.8 ± 0.1 ^cd^	<0.02	3.2 ± 0.4 ^bcd^	3.3 ± 0.4 ^bc^	15.1 ± 1.6 ^a^	1.6 ± 0.1 ^d^
17.6	2-methyldecane	C_11_H_24_	Hydrocarbons	chemical	<0.04	1.3 ± 0.1 ^b^	<0.04	<0.04	<0.04	<0.04	<0.04	<0.04	<0.04	2.1 ± 0.3 ^a^	<0.04	<0.04	<0.04	<0.04	<0.04
17.9	verbenol	C_10_H_16_O	Terpenes	woody	5.8 ± 0.6	<0.02	<0.02	<0.02	<0.02	<0.02	<0.02	<0.02	<0.02	<0.02	<0.02	<0.02	<0.02	<0.02	<0.02
18.4	fenchone	C_10_H_16_O	Terpenes	fruity	3.8 ± 0.2	<0.03	<0.03	<0.03	<0.03	<0.03	<0.03	<0.03	<0.03	<0.03	<0.03	<0.03	<0.03	<0.03	<0.03
18.4	terpinolene	C_10_H_16_	Terpenes	floral	<0.02	<0.02	<0.02	<0.02	<0.02	<0.02	<0.02	<0.02	<0.02	<0.02	<0.02	<0.02	<0.02	3.1 ± 0.4	<0.02
18.7	3-methyl-2-(2methyl-2butenyl furan)	C_10_H_14_O	Furans	floral	<0.04	<0.04	<0.04	<0.04	<0.04	0.3 ± 0.1	<0.04	<0.04	<0.04	<0.04	<0.04	<0.04	<0.04	<0.04	<0.04
18.8	linalool	C_10_H_18_O	Terpenes	floral	4.9 ± 0.5 ^c^	5.1 ± 0.6 ^c^	3.3 ± 0.4 ^d^	9.2 ± 1.0 ^a^	<0.03	3.1 ± 0.4 ^de^	<0.03	1.7 ± 0.1 ^e^	7.4 ± 0.8 ^b^	8.2 ± 0.9 ^ab^	3.6 ± 0.4 ^cd^	4.1 ± 0.5 ^cd^	<0.03	<0.03	<0.03
19.1	1-octen-1ol-acetate	C_10_H_18_O_2_	Esters	fruity	<0.02	<0.02	<0.02	<0.02	<0.02	2.9 ± 0.3 ^b^	<0.02	<0.02	<0.02	<0.02	<0.02	3.6 ± 0.4 ^a^	<0.02	<0.02	<0.02
19.3	thujone	C_10_H_16_O	Terpenes	woody	<0.04	1.1 ± 0.1 ^c^	10.6 ± 1.1 ^a^	<0.04	11.4 ± 1.2 ^a^	<0.04	<0.04	<0.04	<0.04	1.1 ± 0.1 ^c^	4.1 ± 0.5 ^b^	3.5 ± 0.4 ^b^	<0.04	<0.04	<0.04
19.4	cis-4-methoxythujane	C_11_H_20_O	Terpenes	woody	<0.03	<0.03	<0.03	<0.03	<0.03	<0.03	5.4 ± 0.6	<0.03	<0.03	<0.03	<0.03	<0.03	<0.03	<0.03	<0.03
20.2	bornanone	C_10_H_16_O	Ketones	citrus	0.7 ± 0.01 ^c^	<0.04	<0.04	7.0 ± 0.8 ^b^	<0.04	3.1 ± 0.4 ^c^	<0.04	<0.04	<0.04	<0.04	<0.04	<0.04	14.6 ± 1.5 ^a^	<0.04	<0.04
20.6	2-n-pentylthiophene	C₉H₁₄S	Tiofurans	roasted	<0.03	0.6 ± 0.01 ^b^	<0.03	<0.03	<0.03	<0.03	<0.03	<0.03	<0.03	<0.03	6.2 ± 0.6 ^a^	<0.03	<0.03	<0.03	<0.03
20.8	i-menthone	C_10_H_18_O	Terpenes	fruity	<0.02	<0.02	<0.02	<0.02	<0.02	<0.02	<0.02	<0.02	<0.02	<0.02	<0.02	<0.02	3.1 ± 0.3	<0.02	<0.02
21.1	levomenthol	C_10_H_20_O	Terpenes	fruity	<0.03	<0.03	<0.03	<0.03	<0.03	<0.03	<0.03	<0.03	<0.03	<0.03	<0.03	<0.03	3.8 ± 0.4	<0.03	<0.03
21.5	4-Terpineol	C_10_H_18_O	Terpenes	woody	0.6 ± 0.01	<0.05	<0.05	<0.05	<0.05	<0.05	<0.05	<0.05	<0.05	<0.05	<0.05	<0.05	<0.05	<0.05	<0.05
21.7	methyl salicilate	C_8_H_8_O_3_	Esters	floral	<0.04	<0.04	<0.04	<0.04	<0.04	<0.04	<0.04	<0.04	1.9 ± 0.2	<0.04	<0.04	<0.04	<0.04	<0.04	<0.04
22.1	(+)-dihydrocarvone	C_10_H_16_O	Terpenes	floral	<0.05	<0.05	<0.05	<0.05	<0.05	<0.05	4.8 ± 0.5	<0.05	<0.05	<0.05	<0.05	<0.05	<0.05	<0.05	<0.05
23.1	neral	C_10_H_16_O	Aldehydes	fruity	<0.02	<0.02	<0.02	<0.02	<0.02	7.4 ± 0.8	<0.02	<0.02	<0.02	<0.02	<0.02	<0.02	<0.02	<0.02	<0.02
23.2	d-carvone	C_10_H_14_O	Terpenes	floral	<0.02	<0.02	<0.02	<0.02	<0.02	1.5 ± 0.2 ^b^	<0.02	<0.02	<0.02	<0.02	<0.02	<0.02	6.9 ± 0.7 ^a^	<0.02	<0.02
23.5	linalyl acetate	C_12_H_20_O_2_	Esters	floral	28.3 ± 3.1 ^a^	0.8 ± 0.1 ^d^	<0.03	<0.03	<0.03	7.7 ± 0.8 ^c^	<0.03	13.8 ± 1.5 ^b^	4.6 ± 0.5 ^c^	0.4 ± 0.08 ^d^	<0.03	<0.03	<0.03	<0.03	<0.03
24.4	geranyl isovalerat	C_15_H_26_O_2_	Esters	fruity	0.4 ± 0.01	<0.03	1.8 ± 0.1	<0.03	<0.03	7.1 ± 0.7	<0.03	<0.03	<0.03	<0.03	<0.03	<0.03	<0.03	<0.03	<0.03
24.6	medthyl-acetate	C_12_H_22_O_2_	Esters	fruity	<0.04	<0.04	<0.04	<0.04	<0.04	<0.04	<0.04	<0.04	<0.04	<0.04	<0.04	<0.04	2.9 ± 0.3	<0.04	<0.04
24.8	2 3-dimethylhydroquinone	C_8_H_10_O_2_	Ketones	roasted	<0.02	<0.02	<0.02	<0.02	<0.02	<0.02	<0.02	<0.02	<0.02	<0.02	<0.02	3.3 ± 0.4	<0.02	<0.02	<0.02
25.3	thujopsene	C_15_H_24_	Terpenes	floral	<0.03	<0.03	<0.03	<0.03	<0.03	<0.03	<0.03	<0.03	<0.03	<0.03	<0.03	2.7 ± 0.3	<0.03	<0.03	<0.03
26.9	genanyl propionate	C_13_H_22_O_2_	Esters	fruity	<0.03	<0.03	<0.03	<0.03	<0.03	<0.03	<0.03	<0.03	<0.03	0.5 ± 0.01	<0.03	<0.03	<0.03	<0.03	<0.03
27.2	cis α-bergamonete	C_15_H_24_	Hydrocarbons	fruity	<0.05	<0.05	<0.05	2.1 ± 0.3	<0.05	<0.05	<0.05	<0.05	<0.05	<0.05	<0.05	<0.05	<0.05	<0.05	<0.05
28.1	carophyllene	C_15_H_24_	Terpenes	floral	<0.02	<0.02	2.0 ± 0.2 ^cd^	2.2 ± 0.3 ^cd^	5.4 ± 0.6 ^a^	3.9 ± 0.4 ^b^	<0.02	<0.02	<0.02	2.6 ± 0.3 ^c^	<0.02	<0.02	1.8 ± 0.2 ^d^	<0.02	<0.02
28.8	cis-β-farnesene	C_15_H_24_	Hydrocarbons	fruity	<0.04	22.7 ± 2.4 ^a^	<0.04	<0.04	<0.04	<0.04	<0.04	<0.04	<0.04	2.1 ± 0.3 ^b^	<0.04	<0.04	<0.04	<0.04	<0.04
29.5	γ-muurolene	C_15_H_24_	Terpenes	woody	<0.02	<0.02	<0.02	<0.02	<0.02	<0.02	<0.02	<0.02	<0.02	1.2 ± 0.2	<0.02	<0.02	<0.02	<0.02	<0.02
29.7	germacrene	C_15_H_24_	Terpenes	floral	<0.03	3.5 ± 0.4 ^a^	<0.03	2.9 ± 0.4 ^a^	<0.03	<0.03	<0.03	<0.03	<0.03	1.7 ± 0.2 ^b^	<0.03	<0.03	<0.03	<0.03	<0.03
30.6	copaene	C_15_H_24_	Hydrocarbons	woody	<0.03	<0.03	<0.03	<0.03	<0.03	<0.03	<0.03	<0.03	<0.03	0.5 ± 0.01	<0.03	<0.03	<0.03	<0.03	<0.03

Note: Data are expressed as a mean ± standard deviation (n = 3). Values indicated with different letters were significantly different from each other at *p* ≤  0.05 levels, whereas the same letters showed no significant differences (*p* > 0.05). <Below the limit of quantifications.

**Table 2 foods-12-03063-t002:** Content of SFA, MUFA, and PUFA in oil tea samples (%).

Fatty Acid Name	Fatty Acid	T1	T2	T3	T4	T5	T6	T7	T8	T9	T10	T11	T12	T13
caprylic acid	C8: 0	<0.020	<0.020	<0.020	<0.020	<0.020	<0.020	<0.020	<0.020	<0.020	<0.020	<0.020	<0.020	3.46 ± 0.18
capric acid	C10: 0	3.28 ±0.10 ^b^	<0.035	<0.035	<0.035	6.01 ± 0.26 ^a^	<0.035	<0.035	<0.035	<0.035	<0.035	<0.035	<0.035	3.40 ± 0.34 ^b^
lauric acid	C12: 0	<0.020	<0.020	5.16 ± 0.33 ^a^	<0.020	4.86 ± 0.19 ^a^	<0.020	<0.020	<0.020	<0.020	<0.020	<0.020	<0.020	<0.020
myristic acid	C14:0	3.49 ± 0.12 ^cde^	5.18 ± 0.41 ^ab^	5.72 ± 0.25 ^a^	4.69 ± 0.31 ^abcde^	5.66 ± 0.42 ^a^	4.27 ± 0.32 ^bcde^	4.80 ± 0.24 ^abc^	<0.025	4.76 ± 0.27 ^abcd^	3.63 ± 0.22 ^cde^	<0.025	3.41 ± 0.18 ^e^	3.45 ± 0.16 ^de^
myristoleic acid	C14:1 (n-5)	<0.036	<0.036	<0.036	3.06 ± 0.15 ^a^	<0.036	<0.036	<0.036	<0.036	2.42 ± 0.14 ^b^	<0.036	<0.036	<0.036	<0.036
pentadecanoic acid	C15:0	<0.014	<0.014	<0.014	2.43 ± 0.09 ^a^	<0.014	<0.014	<0.014	<0.014	2.48 ± 0.10 ^a^	<0.014	<0.014	1.76 ± 0.08 ^b^	<0.014
cis-10-pentadecanoic acid	C15:1 (n-5)	1.84 ± 0.09	<0.018	<0.018	<0.018	<0.018	<0.018	<0.018	<0.018	<0.018	<0.018	<0.018	<0.018	<0.018
palmitic acid	C16:0	8.89 ± 0.30 ^de^	10.36 ± 1.00 ^de^	16.47 ± 1.10 ^a^	11.80 ± 0.95 ^cd^	11.76 ± 1.10 ^cd^	17.34 ± 1.12 ^a^	15.78 ± 1.10 ^ab^	14.80 ± 1.3 ^abc^	15.09 ± 1.21 ^abc^	10.15 ± 0.98 ^de^	12.22 ± 0.99 ^bcd^	7.20 ± 0.42 ^e^	10.62 ± 1.01 ^de^
palmitoleic acid	C16:1 (n-7)	2.15 ± 0.11 ^def^	2.70 ± 0.11 ^bcd^	3.02 ± 0.18 ^bc^	3.43 ± 0.21 ^b^	2.58 ± 0.14 ^cde^	2.23 ± 0.09 ^cdef^	2.52 ± 0.11 ^cdef^	<0.020	2.60 ± 0.11 ^cde^	1.86 ± 0.08 ^ef^	4.72 ± 0.15 ^a^	1.72 ± 0.07 ^f^	1.90 ± 0.06 ^def^
heptadecanoic acid	C17:0	2.34 ± 0.13 ^cde^	4.45 ± 0.23 ^a^	<0.051	<0.051	2.60 ± 0.11 ^bcd^	2.21 ± 0.10 ^de^	2.53 ± 0.13 ^bcde^	3.05 ± 0.18 ^bc^	2.64 ± 0.19 ^bcd^	2.15 ± 0.16 ^de^	3.28 ± 0.21 ^b^	1.76 ± 0.08 ^e^	1.88 ± 0.07 ^de^
cis-10-heptadecenoic acid	C17:1	2.32 ± 0.11 ^a^	<0.023	<0.023	<0.023	<0.023	<0.023	<0.023	<0.023	<0.023	1.88± 0.18 ^ab^	<0.023	1.74 ± 0.08 ^b^	<0.023
stearic acid	C18:0	6.53 ± 0.41 ^abc^	5.53 ± 0.33 ^bc^	6.86 ± 0.26 ^ab^	6.25 ± 0.24 ^abc^	5.87 ± 0.31 ^bc^	5.90 ± 0.24 ^bc^	6.36 ± 0.26 ^abc^	7.94 ± 0.52 ^a^	5.72 ± 0.32 ^bc^	4.96 ± 0.22 ^c^	6.65 ± 0.35 ^abc^	5.85 ± 0.28 ^bc^	4.96 ± 0.11 ^c^
oleic acid +elaidic acid	C18:1 (cis + trans) (n-9)	8.47 ± 0.32 ^cd^	8.21 ± 0.74 ^cd^	15.28 ± 1.2 ^a^	8.09 ± 0.34 ^cd^	9.01 ± 0.89 ^cd^	7.63 ± 0.51 ^cde^	9.63 ± 0.54 ^c^	9.68 ± 0.72 ^bc^	12.36 ± 1.23 ^b^	6.69 ± 0.42 ^de^	9.55 ± 0.35 ^cd^	5.29 ± 0.32 ^e^	7.08 ± 0.47 ^cde^
linoleic acid + linolelaidic acid	C18:2 (cis + trans) (n-6)	7.33 ± 0.14 ^cdef^	10.07 ± 0.99 ^ab^	9.34 ± 0.32 ^abcd^	6.29 ± 0.33 ^f^	11.52 ± 1.00 ^a^	9.85 ± 0.88 ^abc^	8.78 ± 0.65 ^bcde^	10.14 ± 0.98 ^ab^	9.64 ± 0.65 ^abc^	6.58 ± 0.36 ^ef^	6.97 ± 0.21 ^def^	<0.050	7.78 ± 0.7 ^bcdef^
γ-linolenic acid	C18:3 (n-6)	9.04 ± 0.42 ^a^	6.16 ± 0.29 ^bc^	6.69 ± 0.18 ^b^	<0.031	5.07 ± 0.19 ^bcd^	5.40 ± 0.32 ^bcd^	<0.031	6.36 ± 0.32 ^bc^	4.75 ± 0.33 ^cd^	<0.036	<0.036	3.91 ± 0.20 ^d^	3.83 ± 0.98 ^d^
α-linolenic acid	C18:3 (n-3)	7.70 ± 0.14 ^c^	7.47 ± 0.55 ^c^	7.66 ± 0.51 ^c^	6.81 ± 0.30 ^cd^	4.45 ± 0.15 ^de^	17.15 ± 1.0 ^a^	7.11 ± 0.51 ^cd^	3.88 ± 0.21 ^e^	8.24 ± 0.52 ^c^	15.83 ± 1.02 ^ab^	3.77 ± 0.18 ^e^	6.08 ± 0.21 ^cde^	13.81 ± 0.98 ^b^
arachidic acid	C20:0	8.15 ± 0.25 ^cde^	9.37 ± 0.63 ^bc^	6.61 ± 0.31 ^def^	7.97 ± 0.35 ^cde^	6.06 ± 0.48 ^ef^	19.13 ± 1.10 ^a^	9.15 ± 0.63 ^bcd^	6.31 ± 0.32 ^ef^	5.09 ± 0.32 ^fg^	10.55 ± 1.01 ^bc^	8.32 ± 0.35 ^bcde^	3.16 ± 0.09 ^g^	10.91 ± 0.97 ^b^
gondoic acid	C20:1 (n-9)	2.32 ± 0.14 ^a^	<0.0052	<0.0052	<0.0052	<0.0052	<0.0052	<0.0052	<0.0052	2.51 ± 0.08 ^a^	<0.0052	<0.0052	<0.0052	<0.0052
cis-11,14-eicosadienoic ac	C20:2 (n-6)	4.40 ± 0.21 ^cdef^	3.12 ± 0.22 ^f^	3.71 ± 0.17 ^def^	6.16 ± 0.34 ^cd^	5.81 ± 0.32 ^cde^	2.79 ± 0.12 ^f^	3.28 ± 0.22 ^ef^	14.88 ± 1.10 ^b^	6.33 ± 0.18 ^cd^	2.70 ± 0.11 ^f^	6.75 ± 0.19 ^c^	24.04 ± 1.21 ^a^	2.42 ± 0.09 ^f^
*cis*-8,11,14-Eicosatrienoic acid + heneicosanoic acid	C20:3 (n-6)+ C21:0	3.35 ± 0.18 ^b^	<0.0061	<0.0061	<0.0061	<0.0061	<0.0061	<0.0061	<0.0061	<0.0061	<0.0061	<0.0061	6.37 ± 0.15 ^a^	<0.0061
arachidonic acid	C20:4 (n-6)	3.84 ± 0.14 ^a^	<0.0058	<0.0058	3.03 ± 0.33 ^b^	2.54 ± 0.08 ^b^	<0.0058	<0.0058	<0.0058	<0.0058	<0.0058	<0.0058	<0.0058	<0.0058
cis-11,14,17-eicosatrienoic acid	C20:3 (n-3)	1.68 ± 0.02	<0.0061	<0.0061	<0.0061	<0.0061	<0.0061	<0.0061	<0.0061	<0.0061	<0.0061	<0.0061	<0.0061	<0.0061
eicosadienoic acid	C22:0	1.54 ± 0.08 ^de^	<0.040	<0.040	2.03 ± 0.09 ^cde^	2.10 ± 0.08 ^bcd^	1.78 ± 0.06 ^cde^	<0.040	<0.040	2.21 ± 0.09 ^bc^	1.50 ± 0.01 ^e^	2.68 ± 0.06 ^ab^	1.47 ± 0.09 ^e^	1.45 ± 0.09 ^e^
erucic acid	C22:1 (n-9)	<0.0063	2.46 ± 0.08 ^a^	2.49 ± 0.11 ^a^	<0.0063	<0.0063	<0.0063	<0.0063	<0.0063	<0.0063	<0.0063	<0.0063	<0.0063	<0.0063
cis-4,7,10,13, 16, 19-docosahexanoic acid	C22:2 (n-6)	3.91 ± 0.11 ^e^	6.64 ± 0.32 ^bcd^	4.99 ± 0.18 ^cde^	8.02 ± 0.62 ^b^	9.11 ± 0.62 ^b^	4.34 ± 0.12 ^de^	3.23 ± 0.07 ^e^	17.70 ± 0.98 ^a^	4.26 ± 0.14 ^de^	2.63 ± 0.08 ^e^	16.85 ± 1.1 ^a^	7.63 ± 0.11 ^bc^	2.35 ± 0.05 ^e^
tricosanoic acid	C23:0	5.68 ± 0.21 ^e^	11.79 ± 1.02 ^d^	<0.056	19.93 ± 1.06 ^bc^	<0.056	<0.056	24.80 ± 1.78 ^ab^	<0.056	6.58 ± 0.15 ^de^	26.97 ± 1.20 ^a^	18.23 ± 0.99 ^c^	18.60 ± 1.00 ^c^	20.68 ± 1.33 ^bc^
lignoceric acid	C24:0	1.77 ± 0.09 ^b^	<0.050	<0.050	<0.050	2.19 ± 0.08 ^ab^	<0.050	2.02 ± 0.09 ^ab^	2.46 ± 0.01 ^a^	2.30 ± 0.08 ^ab^	<0.050	<0.050	<0.050	<0.050
cis-4,7,10,13,16,19-docosa-hexanoic + nervonic acid	C22:6 (n-3) + C24:1 (n-9)	<0.0086	6.50 ± 0.31 ^a^	6.00 ± 0.41 ^a^	<0.0086	<0.0086	<0.0086	<0.0086	<0.0086	<0.0086	<0.0086	<0.0086	<0.0086	<0.0086
**Σ SFA**		45.0 ± 3.2 ^cde^	46.7 ± 3.2 ^bcde^	40.8 ± 2.8 ^de^	55.1 ± 4.12 ^abcd^	47.1 ± 3.25 ^bcde^	50.6 ± 3.21 ^abcde^	65.5 ± 4.23 ^a^	37.4 ± 2.11 ^e^	46.9 ± 3.2 ^bcde^	59.9 ± 3.1 ^abc^	51.4 ± 2.8 ^abcde^	49.6 ± 1.89 ^bcde^	60.8 ± 1.8 ^ab^
**Σ MUFA**		17.1 ± 1.10 ^bc^	19.9 ± 1.20 ^b^	26.8 ± 1.86 ^a^	14.6 ± 1.00 ^cd^	11.6 ± 0.98 ^def^	9.9 ± 0.61 ^ef^	12.1 ± 0.98 ^def^	9.7 ± 0.31 ^f^	19.9 ± 1.00 ^b^	10.4 ± 0.99 ^def^	14.3 ± 0.98 ^cde^	8.8 ± 0.24 ^f^	9.0 ± 0.4 ^f^
**Σ PUFA**		37.9 ± 2.87 ^bc^	39.9 ± 2.10 ^bc^	38.4 ± 2.64 ^bc^	30.3 ± 2.65 ^cd^	41.3 ± 3.21 ^b^	39.5 ± 1.25 ^bc^	22.4 ± 1.12 ^d^	53.0 ± 3.21 ^a^	33.2 ± 0.65 ^bcd^	29.7 ± 1.02 ^cd^	34.3 ± 1.25 ^bc^	41.7 ± 2.32 ^b^	30.2 ± 1.6 ^cd^
**PUFA/MUFA**		2.22 ± 0.10 ^efg^	2.01 ± 0.14 ^efg^	1.43 ± 0.05 ^g^	2.08 ± 0.08 ^efg^	3.57 ± 0.18 ^cd^	4.01 ± 0.11 ^bc^	1.84 ± 0.06 ^fg^	5.47 ± 0.22 ^a^	1.67 ± 0.02 ^fg^	2.85 ± 0.08 ^de^	2.41 ± 0.07 ^ef^	4.76 ± 0.22 ^ab^	3.36 ± 0.1 ^cd^
**Σ n-6 PUFA**		28.51 ± 1.78 ^bcde^	25.98 ± 1.89 ^cde^	24.74 ± 1.85 ^de^	23.49 ± 1.28 ^ef^	34.06 ± 2.15 ^bc^	22.38 ± 0.89 ^ef^	15.29 ± 1.13 ^fg^	49.08 ± 2.14 ^a^	24.98 ± 1.88 ^de^	11.91 ± 0.99 ^g^	30.57 ± 1.85 ^bcd^	35.58 ± 2.33 ^b^	16.39 ± 0.9 ^fg^
**Σ n-3 PUFA**		12.73 ± 0.9 ^abc^	13.97 ± 1.02 ^ab^	13.65 ± 1.03 ^ab^	6.81 ± 0.25 ^d^	7.26 ± 0.26 ^cd^	17.15 ± 1.10 ^a^	7.11 ± 0.11 ^d^	3.88 ± 0.08 ^d^	8.24 ± 0.21 ^bcd^	17.76 ± 0.99 ^a^	3.77 ± 0.08 ^d^	12.46 ± 0.95 ^d^	13.81 ± 1.0 ^a^

Data are expressed as percentages of total FAMEs and represent the mean ± standard deviation (n = 3). The values with different superscript letters are significantly different (*p* < 0.05).

**Table 3 foods-12-03063-t003:** Minerals, nitrogen, and content of polyphenols for studied tea samples expressed as averages ± standard deviation (n = 3).

Tea Samples		Major Elements					Trace Elements			Polyphenol g GA/kg
	Na mg/kg	K mg/kg	Ca mg/kg	Mg mg/kg	N mg/kg	P mg/kg	Fe mg/kg	Cu mg/kg	Zn mg/kg	
**T1**	24.8 ± 1.9 ^c^	5661 ± 71 ^f^	3610 ± 42 ^gh^	980 ± 20 ^e^	2.35 ± 0.11 ^e^	347 ± 28 ^ij^	10.9 ± 0.9 ^gh^	1.35 ± 0.12 ^f^	4.30 ± 0.36 ^gh^	19.3 ± 2.0 ^fg^
**T2**	669 ± 35 ^a^	30,400 ± 717 ^a^	11,595 ± 366 ^bc^	2415 ± 40 ^bc^	5.56 ± 0.20 ^a^	5545 ± 229 ^a^	53.7 ± 4.9 ^ef^	5.41 ± 0.50 ^bcd^	42.6 ± 3.94 ^a^	9.5 ± 1.0 ^g^
**T3**	71.6 ± 3.6 ^c^	4954 ± 173 ^fg^	10,935 ± 215 ^c^	1646 ± 56 ^de^	4.01 ± 0.13 ^bcd^	1117 ± 41 ^hi^	157 ± 12 ^b^	1.10 ± 0.10 ^fg^	12.2 ± 1.1 ^ef^	28.3 ± 2.9 ^ef^
**T4**	36.5 ± 3.5 ^c^	15,036 ± 131 ^cd^	19,504 ± 474 ^a^	3036 ± 84 ^ab^	3.63 ± 0.13 ^cde^	2035 ± 81 ^efg^	391 ± 15 ^a^	4.15 ± 0.39 ^de^	41.7 ± 3.87 ^a^	53.3 ± 5.4 ^bc^
**T5**	34.0 ± 3.2 ^c^	20,546 ± 342 ^b^	7632 ± 108 ^def^	2504 ± 68 ^bc^	5.11 ± 0.17 ^ab^	2079 ± 69 ^efg^	43.7 ± 3.4 ^efg^	6.79 ± 0.67 ^ab^	28.7 ± 2.4 ^b^	23.6 ± 2.4 ^efg^
**T6**	43.6 ± 1.5 ^c^	28,376 ± 294 ^a^	11,243 ± 122 ^bc^	2909 ± 84 ^ab^	5.17 ± 0.30 ^ab^	3365 ± 132 ^c^	75.5 ± 5.5 ^de^	4.70 ± 0.42 ^cde^	21.3 ± 1.6 ^cd^	62.6 ± 6.4 ^bv^
**T7**	50.4 ± 4.0 ^c^	17,230 ± 364 ^bc^	6842 ± 84 ^ef^	1725 ± 120 ^d^	4.79 ± 0.20 ^abc^	2387 ± 56 ^def^	134 ± 11 ^bc^	6.45 ± 0.64 ^ab^	11.2 ± 0.95 ^efg^	86.8 ± 8.8 ^a^
**T8**	65.8 ± 6.6 ^c^	16,289 ± 370 ^bcd^	2588 ± 92 ^h^	1315 ± 38 ^de^	5.52 ± 0.23 ^a^	3111 ± 53 ^cd^	75.4 ± 6.8 ^de^	7.70 ± 0.74 ^a^	19.9 ± 1.76 ^d^	18.9 ± 1.9 ^fg^
**T9**	55.4 ± 1.5 ^c^	21,190 ± 286 ^b^	5635 ± 75 ^fg^	3248 ± 60 ^a^	5.30 ± 0.20 ^a^	4528 ± 108 ^b^	44.3 ± 3.4 ^efg^	6.75 ± 0.63 ^ab^	17.7 ± 1.3 ^de^	44.5 ± 4.5 ^cd^
**T10**	54.9 ± 3.0 ^c^	12,350 ± 110 ^cde^	10,170 ± 198 ^cd^	2925 ± 62 ^ab^	4.83 ± 0.22 ^abc^	2601 ± 40 ^cde^	63.7 ± 3.4 ^e^	5.90 ± 0.51 ^bc^	27.5 ± 2.29 ^bc^	44.7 ± 4.6 ^cd^
**T11**	43.1 ± 2.5 ^c^	13,260 ± 205 ^cde^	1484 ± 20 ^h^	1872 ± 36 ^cd^	4.47 ± 0.23 ^abc^	1707 ± 45 ^fgh^	23.7 ± 2.2 ^fg^	1.20 ± 0.09 ^f^	8.91 ± 0.76 ^fg^	34.7 ± 3.5 ^de^
**T12**	40.8 ± 3.3 ^c^	9527 ± 104 ^ef^	13,930 ± 69 ^b^	2442 ± 37 ^bc^	2.76 ± 0.14 ^de^	1541 ± 35 ^gh^	37.3 ± 1.6 ^efg^	1.75 ± 0.13 ^f^	17.6 ± 1.65 ^de^	61.8 ± 6.3 ^b^
**T13**	290 ± 16 ^b^	12,130 ± 175 ^de^	9230 ± 128 ^cde^	3345 ± 73 ^a^	4.58 ± 0.21 ^abc^	2522 ± 43 ^de^	114 ± 10 ^cd^	3.65 ± 0.27 ^e^	7.90 ± 0.64 ^fg^	85.1 ± 8.6 ^a^
**T14**	33.2 ± 2.7 ^c^	122 ± 8 ^gh^	1391 ± 59 ^h^	26.5 ± 2.2 ^f^	0.27 ± 0.02 ^f^	286 ± 8 ^j^	<1.66	<0.66	<0.66	12.2 ± 1.3 ^gh^
**T15**	62.0 ± 4.4 ^c^	29.3 ± 2.1 ^h^	1251 ± 43 ^h^	11.9 ± 1.1 ^f^	0.31 ± 0.02 ^f^	346 ± 11 ^ij^	<1.66	<0.66	<0.66	15.2 ± 1.6 ^fg^

Note: Values indicated with different letters were significantly different from each other at *p* ≤ 0.05 levels, whereas the same letters showed no significant differences (*p* > 0.05). Different letters in each column showed a significant difference at the level of *p* ≤ 0.05.

## Data Availability

The data discussed within this study is available from the corresponding author upon request.

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
