# Peer review of "Chemical Analysis of Various Tea Samples Concerning Volatile Compounds, Fatty Acids, Minerals and Assessment of Their Thermal Behavior"

_foods, 2023, doi:10.3390/foods12163063_

Round 1

Reviewer 1 Report

An interesting formula and I do enjoy reading this type of research. However, it suffers from various critical issues as outlined below:

a)      The purpose of the paper is not clear. The method by which the samples were chosen does not reflect what was stated in the introduction, in which the cited bibliography refers only to tea (see references 1 -11), while no articles referring to the other plants under study have been cited. Furthermore, thirteen samples of tea made from different plants were compared with only two samples of tea produced from Camellia sinensis. I can't find a possible comparison between the two groups.

b)      Oil extraction from teas: since a derivatization reaction was performed, the use of an internal standard would have been appropriate.

c)       Line 144 Please clarify: Each oil sample was trimethylated and analyzed in three replicates.

d)      Line 154 standard mixtures are repeated twice. Please correct.

e)      Volatile composition: even in the case of using the SMPE technique, it would have been appropriate to use an internal standard. Furthermore, the method used for the interpretation of the mass spectra of the different compounds is completely missing. Please add how the interpretation was done. It would also have been appropriate to use the Kovats retention indices, a technique that allows the recognition of volatile compounds in support of the mass spectrum.

f)       Table 1: if the analyzes were done in triplicate, specify in the caption (mean + standard deviation).

g)      Antioxidant characterization: The Folin-Ciocalteu test refers only to the amount of total polyphenols and not to the antioxidant activity. If you want to talk about antioxidant activity, after the Folin test it would be appropriate to add at least the DPPH test.

h)      Line 287 Please clarify: Eucalyptol was the most abundant group of volatiles present in the tea samples.

i)        Results and Conclusions must be implemented

A thorough revision of the English language is required.

Reviewer 2 Report

The introduction is comprehensive, with all necessary data and adequate literature. It ends with a clearly set goal of the experiment. But check lines 72-73 and reference 13: …omega 3 FA can cause stroke, age-related cognitive decline…” Is this correct?

The methodology is comprehensive with a large number of samples and clearly defined test methods that are reproducible.

There are many results and they are well presented with the addition of multivariate analysis. Precisely this large amount of data, summarized in one manuscript, can be used for comparison for further researches. However, what is lacking in this work is the insufficient connection of the obtained results with previous researches. It is necessary to find more researches/results and supplement the discussion in order to give the paper scientific significance. Suggestion for authors is to expand the discussion by comparing their results with more previous researches/results in this scientific field. This is why I suggest minor revision.

The conclusion is clear and follows from the obtained research results.

Minor editing of English language required

Reviewer 3 Report

Line 41-43: this sentence needs to be removed or rewritten. It has nothing to do with the paragraph.

 Line 191 and Line 199: More detailed information about the methods should be given. In addition, information about statistical analysis should also be given. The study has no a experimental design. It is unclear how many times samples were taken. There is no information on how the sampling was done.

Line 211:Figures should be given after explanations.

Many compounds can be found in the analysis of volatile compounds. However, verification/identification is required. How identification was performed in this study is not specified.

On the other hand, I suggest reevaluation of terpene compounds by separating them from hydrocarbons and alcohols.

There is no statistical evaluation in the tables or in the manuscript. Differences cannot be mentioned without statistical analysis.

Reviewer 4 Report

The manuscript deals with the Chemical analysis of various tea samples concerning volatile compounds, fatty acids, minerals, and assessment of their thermal behavior. I find this study on tea leaves, flowers, and fruits fascinating! It's intriguing to see how thermal decomposition was tracked, leading to a better understanding of the tea components' composition. The abundance of polyphenols in tea is impressive and could explain its well-known antioxidant properties.

Overall, the topic is interesting and catches the audience’s attention, but it needs to improve according to the suggested comments.

Abstract:

L 15 to 17: From the thermal……to lignin content. Rewrite this sentence with more clarity   

It appears that the text is missing appropriate concluding sentences.

Introduction:

Lines 32 and 33: Add some possible statistics regarding production and consumption with proper reference.

L 96 to 98: Polyphenols of 20-30 % of the dry weight of tea leaves are the most abun- 96 dant class of soluble components that influence the color, taste, aroma of tea leaves and 97 the most important substance to exert its health benefits. Please specify the type of tea.

L 101 to 107: Please clarify the last paragraph.

The lacking part of the introduction is a clear statement of the research gap or need for the current study. While the introduction mentions several studies but does not explicitly state the specific gap or problem the current study aims to address. In addition, there is no direct link between the previous studies and the objectives of the current study.

Material and methods:

Line 196 to 197: How to determine lignin?

Results and discussion

L 351 to 352: We categorized precursor molecules into five groups: carotenoids, fatty acids, glycosides, amino acids/carbohydrates, and other precursors, including isoprenoids, polyphenols, and unknowns. What was the reason for this categorization?

Line 371: Data are expressed as a percentage of total FAMEs and represent mean ± standard deviation (n = 3): Write a detailed caption for Table 2.

Conclusion:

Need to add a few concluding sentences

Minor editing of English language required.

Round 2

Reviewer 1 Report

Thanks for completing the suggested fixes.

Now the work is very clear and complete.

Best regards

Reviewer 3 Report

Authors have made the necessary explanations and corrections. It can be accepted for publication.